# Physicochemical Properties and Bioactivity of a New Guar Gum-Based Film Incorporated with Citral to Brown Planthopper, *Nilaparvata lugens* (Stål) (Hemiptera: Delphacidae)

**DOI:** 10.3390/molecules25092044

**Published:** 2020-04-28

**Authors:** Xiubing Gao, Can Guo, Ming Li, Rongyu Li, Xiaomao Wu, Anlong Hu, Xianfeng Hu, Feixu Mo, Shuai Wu

**Affiliations:** 1Institute of Plant Protection, College of Agriculture, Guizhou University, Guiyang 550025, Guizhou, China; 2Guizhou Tea Research Institute, Guizhou Province Academy of Agricultural Science, Xiaohe District, Guiyang 550006, Guizhou, China; 3The Provincial Key Laboratory for Agricultural Pest Management in Mountainous Region, Guizhou University, Guiyang 550025, Guizhou, China

**Keywords:** *Nilaparvata lugens*, film, citral, guar gum, physicochemical properties, bioactivity

## Abstract

The brown planthopper (BPH), Nilaparvata lugens (Stål), is the most notorious rice insect pest. In order to repel BPH effectively while being environmentally friendly, a new film based on guar gum incorporated with citral (GC film) was formulated. A toxicity bioassay of citral and guar gum at different proportions (ratios of 3:1, 2:1, 1:1, 1:2, and 1:3 in *w*/*w*) of GC film-forming emulsion to BPH was performed with the rice stem dipping method. Results showed that the most effective ratio of citral to guar gum was 1:1 with the median lethal concentration (LC_50_) of 4.30 mg/mL, far below the LC_50_ of guar gum (GG)/citral individual (141.51 and 44.38 mg/mL, respectively). The mortality of BPH adults and nymphs in the third instar treated with different dilution multiples of GC film-forming emulsion ranged from 46.67% to 82.22% and from 37.78% to 71.11%, respectively. These indicated that GC film-forming emulsion had a direct toxicity on BPH, and the mixture of citral and GG had synergistic interactions. Subsequently, Fourier-transform infrared spectroscopy showed that the incorporation of guar gum with citral was successful and did not result in the formation of new chemical bonds. The GC film exhibited a darker color and rougher surface topography with larger apertures and deeper gullies (Ra = 1.42 nm, Rq = 2.05 nm, and Rmax = 25.40 nm) compared to the guar gum film (GG film) (Ra = 1.00 nm, Rq = 1.33 nm, and Rmax = 16.40 nm), as determined by transmission electron microscopy and atomic force microscopy. The GC film exhibited a 50.4% lower solubility in water (30.30% vs. 15.00%) and 71.3% oxygen permeability (8.26 × 10^−9^ vs. 2.37 × 10^−9^ cm^3^/m^2^·d·Pa) (*p <* 0.05) but did not demonstrate any significant difference in mechanical properties, such as thickness (39.10 vs. 41.70 mm), tensile strength (41.89 vs. 38.30 N/mm^2^), and elongation at break (1.82% vs. 2.03%) (*p <* 0.05) compared to the GG film. Our findings established a link between physicochemical properties and bioactivity, which can provide useful information on developing and improving GC films and may offer an alternative approach for the control of BPH in the near future.

## 1. Introduction

Rice (*Oryza sativa* L.) is one of the world’s most important food crops. It is the primary source of calories for more than one-third of the global population and a staple food for over half of the population in Asian countries. Diseases and insect pests are the major constraints to rice production throughout rice-growing countries. The brown planthopper (BPH, *Nilaparvata lugens* Stål, Hemiptera: Delphacidae) is the most notorious insect pest of rice. In recent years, BPH has caused devastating damage to rice crop in Asian countries. In 2005 and 2008, a combined yield loss of 2.7 million tons of rice was reported in China, while the yield loss was 0.4 million tons in Vietnam [1], due to direct damage by BPH. BPH causes direct damage to the rice plants by sucking the phloem and infecting the plant with ragged stunt virus diseases, particularly grassy stunt virus (RGSV) and ragged stunt virus (RRSV) [1].

Insecticides are considered the most effective and frequently used approach to control BPH in modern agriculture [2,3,4]; however, they are costly in terms of labor, cost, and environment. More seriously, BPH has evolved different levels of resistance to many major classes of insecticides due to the heavy and frequent application of insecticides [4,5,6,7,8]. According to the latest update of the Arthropod Pesticide Resistance Database (APRD), a total of 419 cases of insecticide resistance have been reported throughout the world [9]. Therefore, alternative and environmentally friendly methods and materials to control BPH are needed. Recently, the development of bio-based active films to reduce the use of chemical pesticides has received much interest [10]. For example, Peng et al. (2011) reported a useful alternative for the management of potato psyllids, *Bactericera cockerelli* (Hemiptera: Psyllidae) by kaolin particle films [11]. Giteru et al. (2015) developed a kafirin-based bioactive film incorporated with citral that exhibits antimicrobial activities against *Campylobacter jejuni, Listeria monocytogenes,* and *Pseudomonas fluorescens* [12]. Sahraee et al. (2017) presented a gelatin-based nanocomposite film with anti-fungal properties against *Aspergillus niger* in the contact surface zone [13]. These bio-based active films also possess the ability to avoid moisture loss or water absorption, oxygen penetration, aroma loss, and solute transport [14]. Although films with antimicrobial activities have been extensively studied, the research on films repelling injurious insects is scarce.

Active films are usually based on water-soluble natural film-forming materials, such as polysaccharides, plant proteins, and high-molecular polymers. A great number of works based on the preparation of films by using galactomannans from various sources have been reported [15,16,17,18]. Guar gum (GG) is a naturally occurring water-soluble polysaccharide that can be obtained from the seed of the legume *Cyamopsis tetragonalobus*. It is also a functional galactomannan that consists of mannose units linked via β-1,4-linkage as the backbone and galactose units linked via α-1,6-linkage in the side chain [19]. GG is known to have one of the highest molecular weights (Mw), in the vicinity of ~2.8 × 10^7^. Given that the viscosity of GG is large due to a high mannose/galactose ratio, it can easily form a stable and strong film on the surface of plants [18,20]. Vilas et al. (2018) reported a tri-phasic edible film from GG loaded with plant-based antimicrobial compounds [21]. Saberi et al. (2017a, 2017b) presented an edible pea starch-GG biocomposite edible film as an antimicrobial agent for active food packaging [22,23]. Dhumal et al. (2019a, 2019b) reported that GG films with essential oils as active packaging materials have the potential to function as antimicrobial agents [24,25].

To enhance bioactivity (e.g., antimicrobial and antioxidant activity), plant essential oils and natural antioxidants (polyphenols) have been incorporated into bio-based films. For example, essential oils from oregano, rosemary, and garlic were incorporated into whey protein films [26], and cinnamon essential oils were incorporated into chitosan-based films to improve their antimicrobial and physicochemical properties [27]. The incorporation of plant essential oils and polyphenols into bioactive films might be a feasible approach for improving their industrial applications. Essential oil has been reported to be effective in inhibiting the growth of a wide range of microorganisms [28].

Citral, an open-chain monoterpene aldehyde present in the essential oils of several medicinal plants, including *Backhousia citriodora* F. Muell (Myrtaceae) (90–98%), *Ocimum basilicum* L.(Lamiaceae) (55–99%), *Cymbopogon citratus* (Grasses) (65–80%), and *Citrus sinensis* Osbeck (Rutaceae) (0.7–3%) [29], is the major compound of citrus-derived essential oils. Citral has not only antifungal and bactericidal activities [30,31], but also insecticidal activity [32,33,34,35]. Unfortunately, citral is susceptible to oxidative degradation and unstable in water under neutral pH, and, as a result, citral easily loses its biological activity under normal storage conditions. Thus, exploring a new approach to improve the toxicity of citral is key to enhancing the efficacy of this plant essential oil [35].

However, the application of these functional films to repel BPH has not been documented thus far. Additionally, little attention has been paid to the importance of the effect of incorporating plant essential oil on the physical and mechanical properties of these films. At this point, we have developed a film based on the excellent film-formed material GG and incorporated the bioactive essential oil citral to effectively repel BPH and be environmentally friendly. In the present study, the effect of this film on the third instar nymph of BPH was investigated using the rice stem dipping method. Furthermore, its physical and mechanical properties were also characterized through scanning electron microscopy (SEM) and atomic force microscopy (AFM), respectively. The obtained results are expected to provide a new approach for the controlling of BPH and useful information on the internal relationship among the film’s physical properties, the microstructure of the film-forming solution, and the resultant film microstructure.

## 2. **Results**

### 2.1. Bioactivity of Film-forming Emulsion to BPH

The median lethal concentration (LC_50_) values of different ratios of citral and GG emulsion of the film (GC film), citral and GG to BPH, are shown in Table 1. Among all the constituents tested, the ratio 1:1 of citral and GG emulsion was the most toxic and had an LC_50_ of 4.30 mg/mL, whereas the lowest toxicity was shown by the ratio 1:3, with an LC_50_ of 15.43 mg/mL. The results showed that the GC film with a 1:1 GG–citral ratio had the optimum composition of bioactivity film to repel BPH. The average LC_50_ value of different ratios of citral and GG emulsion of the GC film was 8.14 mg/mL, far below the LC_50_ of GG/citral individual (141.51 and 44.38 mg/mL, respectively), showing that the mixture of different ratios of citral and GG had synergistic interactions.

Based on this optimum composition, the GC film-forming emulsion and GG film-forming emulsion were prepared, and their effects on the mortality of BPH are shown in Figure 1. The GC film-forming emulsion had a remarkable effect on the mortality of BPH (46.67% to 82.22% and 37.78% to 71.11% on BPH adults and nymphs in the third instar, respectively), and this effect positively correlated with the treatment concentration, as particularly observed in the BPH adults. The GG film-forming emulsion had little effect on the mortality of BPH when treated at high concentrations (100 and 200 dilution multiples on BPH adults’ mortality were 15.56% and 13.33%, respectively, and 100 dilution multiples on BPH nymphs’ mortality was 11.11%), but this effect was not significant when the concentration was reduced compared to the control. These showed that GC film-forming emulsion had a direct toxicity on BPH, as the GG was the main film-forming material and could have a repellent or antifeedant effect on BPH.

### 2.2. FTIR Spectroscopy Analysis

FTIR spectroscopy was performed to study the changes in chemical structures. As shown in Figure 2A–C, the absorption broadband at 3452 cm^−1^ indicates the presence of O–H stretching vibrations of the gum polysaccharide, which arose in all the films. The strong absorption peaks of the GC film-forming emulsion (Figure 2A) showed no obvious differences compared to pure GG (Figure 2C), except for the bands at 1649 cm^−1^. The peak at 1649 cm^–1^ showed weak absorption and was the characteristic absorption peak of citral. By contrast, the strong absorption peaks of the GG film-forming emulsion (Figure 2B) were significantly changed at 2902 and 2995 cm^−1^, which are assigned to C–H and O–H stretching vibrations, respectively.

### 2.3. Microstructure of the Films

As shown in Figure 3A,B, all of the films were flexible in nature. The *GC film* was light yellowish in color, and the *GG film* was whitish. Figure 4A–D, a and b show that all of the surfaces of the films were smooth and had no cracks. The surface of the GC film was rougher and had larger apertures and deeper gullies (Ra = 1.42 nm, Rq = 2.05 nm, and Rmax = 25.40 nm) (Figure 5C,D) but fewer nodules (Figure 5A,B), compared to those of the GG film (Ra = 1.00 nm, Rq = 1.33 nm, and Rmax = 16.40 nm). These micrographs clearly show the changed surface morphology of GG after adding citral.

### 2.4. Physicochemical Properties of the Films

Thickness, TS, and elongation at break (EAB) were used to describe the mechanical properties of the films. As shown in Table 2, the GC film had the same thickness (39.10 vs. 41.70 mm), TS (41.89 vs. 38.30 N/mm^2^), and EAB (1.82% vs. 2.03%) as the GG film (*p <* 0.05). This result indicates that as the forming material, GG determined the mechanical properties of the film. The films incorporated with citral showed significant changes in physical properties. The solubility in water (SOW) (30.30% vs. 15.00%) and oxygen permeability (OP) (8.26 × 10^−9^ vs. 2.37 × 10^−9^ cm^3^/m^2^·d·Pa) of the GC film were lower than those of the GG film (*p <* 0.05), reduced by approximately 50.4% and 71.3%, respectively, suggesting that the physical properties of the films were deeply affected by the incorporation of citral essential oil.

## 3. Discussion

During the past decades, there has been an increasing interest in finding new and reliable sources of insecticides to reduce the use of chemical pesticides. Although the development of new insecticides is still of great significance, the studies on improving the efficacy of those insecticides already in use (e.g., neem and essential oils) deserve greater attention because the latter may present fewer regulation hurdles. Citral displays excellent antimicrobial activity against pathogens [30,36] and pests [32,33,34]. However, citral is susceptible to oxidative degradation, which results in the loss of activity under normal storage conditions. Therefore, finding a way to improve citral’s toxicity and make it stable could be the key to enhancing the efficacy of citral. Up until now, a number of citral-based antimicrobial packaging films have been reported in the literature [12,37]. The finding of the present investigation reveals the repelling effect of the GC film on BPH and its physics characterization.

The effectiveness of the GC film in repelling BPH was proven via the rice stem dipping method and spraying method. The GC film with a GG–citral ratio of 1:1 (*w*/*w*) was the optimum composition of the bioactivity film to repel BPH, with an LC_50_ of 4.30 mg/mL, which is far below the LC_50_ of the GG/citral individual, showing that the mixture of citral and GG had synergistic interactions. This was an unexpected result for two reasons. On the one hand, the above-presented results indicate a valuable and promising potential of the GC film for rice diseases and pest control, as citral exhibited excellent antimicrobial activity against *Pyricularia grisea* (EC_50_ = 39.52 ± 7.70 ug/mL) [38], which is one of most important rice diseases worldwide. On the other hand, the LC_50_ value of the GC film BPH was 4.30 mg/mL, which is in line with that of methyl eugenol, a phenylpropanoid, with an LC_50_ value of 1.025 mg/mL to BPH [39], but much smaller than those of commercial biopesticides (the LC_50_ of botanical pesticides ranged from 0.14 to 110.75 μg/mL, microbial pesticides ranged from 403.90 to 728.29 μg/mL, and agricultural antibiotics was about 0.25 μg/mL) [40] and synthetic insecticides (average LC_50_ ranged from 3.54 to 49.85 ug/mL) [41]. Thus, the effect of the GC film on repelling BPH is limited and still needs further research.

The mortality of BPH treated with different concentrations of GC film-forming emulsion ranged from 46.67% to 82.22% and from 37.78% to 71.11% in BPH adults and nymphs in the third instar, respectively. These results showed GC film-forming emulsion had a significant effect on the mortality of BPH. Compared to the GC film-forming emulsion, the GG film-forming emulsion had less effect on the mortality of BPH treated with high concentrations only. Therefore, it was suggested that GC film-forming emulsion had a direct toxicity on BPH, and citral is the major active component of GC film-forming emulsion to repel BPH. GG film-forming emulsion also had little effect on BPH maybe due to the excellent film-forming property of GG, which could form a barrier on the surface of rice and affect the behavior of BPH, such as recognition, probing, and feeding. Puterka et al. (2000) reported that plants coated with a hydrophobic particle film barrier become visually or tactilely unrecognizable as a host [42]. In addition, insect movement, feeding, oviposition, and other activities can also be severely impaired [43]. However, the mechanism is still unclear and requires further study.

To the best of our knowledge, the effectiveness of citral and GG, individually or in combination, in repelling BPH has never been reported. Therefore, this study is the first to report a GG-based film incorporated with citral that has bioactivity to BPH. Our film is effective in repelling BPH probably because citral has been reported to have a certain cytotoxicity in the ovarian cell line of several insects [35,44,45], and a mixture of GG and Tween-20 coatings enhanced the physicochemical stabilization of citral and protected it from degradation [36,37,46,47,48].

The chemical structures of the GC film-forming emulsion, GG film-forming emulsion, and pure GG films were compared by FTIR. The broad peak at 3452 cm^−1^ appeared in each sample, indicating the presence of O–H stretching vibrations of gum polysaccharide [49]. The peak at 1649 cm^−1^, which showed a weak absorption, was assigned to the stretching vibration of –C=O in a carbonyl group and was the characteristic absorption peak of citral [49]. These characteristic absorption peaks showed that the incorporation of the GG with citral was successful and did not result in the formation of new chemical bonds. The weak absorption of these bands was likely due to fact that citral was coated by GG; hence, its molecular vibration was limited and it could not reveal its original infrared characteristics [50]. We, therefore, speculated that the citral coating the guar gum in the GC film was more non-volatile and not readily susceptible to oxidation, which greatly improved the activity repelling BPH. The broad peaks ranging from 2902 and 2995 cm^−1^ were assigned to the C–H stretching and to the O–H stretching vibrations, respectively [16]. These peaks did not appear in the GG film-forming emulsion. The peaks at 2902 and 2995 cm^−1^ were associated with the presence of asymmetric and symmetric methylene stretching vibrations of tween 20, respectively [51], so we can infer that the addition of tween 20 altered the C–H stretching and O–H stretching vibrations in the emulsion.

The mechanical properties (thickness, TS, and EAB) of the GC and GG films were almost the same, except for coloration. The GC film passed on a light-yellow color, whereas the GG film showed a whitish shade of color. Kim et al. (1995) reported that citral passes on a yellowish coloration to citral-treated fish products [52]. Thus, we speculated that the mechanical properties of the two films mainly depended on the film-formed material GG, and coloration mainly depended on citral.

The physical properties of the GC and GG films, such as SOW and OP, were significantly different. The reduction in solubility of the GC film could be due to consequences of the enhanced interaction of GG and citral, which might lead to a more compressed structure and less SOW. These results are consistent with previous reports [25,52,53,54]. In general, film solubility depends greatly on the hydrophobic and hydrophilic index values of the various components of the films [55]. The reduction in the SOW of the essential oil-impregnated films is attributed to their extreme lipophilic nature [12], which increases the water resistance of the films [56].

Oxygen as a non-polar molecule tends to dissolve in low-polarity polymers [57]. The OP of the films in this study mainly depended on the tightness of their components [58,59]. The low OP shown by the GC film indicates that incorporation of citral could have remarkably reduced the film’s OP. The reason could be the highly intimate interaction between the citral and the GG, which resulted in the reduction diffusion of oxidizing molecules across the film [60]. The results of our study indicate that the incorporation of citral could significantly reduce the film’s OP.

The surface of the GC film was rougher than that of the GG film. As a general trend, the microstructure of the films made from the oil films was rougher than that of oil-free films [61]. The increase in surface coarseness with the presence of essential oils has been previously observed by Norajit et al. (2010) and Shojaee-Aliabadi et al. (2014) [61,62], who attributed this fact to the migration of oil droplets in the upward direction of the films and further volatilization during water evaporation, resulting in a holey structure [63]. The rough surface of the GC film in the current study was conducive to repelling BPH, providing a barrier. This point was much similar to particle films, which can affect insect behavior through contact with treated surfaces [43,64].

Today, because alternative pest management tactics are needed to help make crop protection more sustainable, the European-level Regulation (EC) No. 1107/2009 [65] encouraged the development of less harmful substances [66]. Natural products, such as biocidal plant extracts, are one of the main integrated pest management (IPM) tactics and may play an important role in the implementation of the IPM [66,67]. The GC film comprised plant extracts and plant-derived substances and has been shown to repel BPH effectively, so it was supposed to be an application potential and operable technology to make IPM work in rice production. Furthermore, the GC film may not be limited to only repelling BPH or species of planthoppers, but also other pierce-sucking insect pests for a range of attractive physicochemical properties via novel mechanisms of action. Thus, the data we obtained in our study established a link between physicochemical properties and bioactivity and could provide useful information on developing and improving GC films to repel other insect pests.

## 4. Materials and Methods

### 4.1. Chemicals

The 97% pure citral used in this study was isolated and extracted from the essential oils of *Litsea cubeba*, mixed with cis- and trans- isomers, and stored at 4 °C. GG was of food grade and in the local packaging imported, originally procured from Shree Ram Gum Chemicals Pvt. Ltd., Rajasthan, India. Polyoxyethylene (20) sorbitan monolaurate (Tween 20, hydrophilic–lipophilic balance = 16.7) was purchased from Shanghai Aladdin Biochemical Technology Co., Ltd. (Shanghai, China). The water used in this study was deionized and filtered using an ultra-pure water purifier, namely, Molecular Molatom 18,100 (Molecular Instrument Shanghai Co., Ltd., Shanghai, China).

### 4.2. Rice and Insects

The three-line hybrid rice plant, Jinyou 785, which is susceptible to BPH, was utilized to raise BPH and for the experiments described below. BPH was originally obtained from a rice paddy in Jiuzhou Town, Huangping County, Guizhou Province, China (26°99’ N, 107°74’ E), with more than 1000 nymphs and adults being caught in June, 2016. These insects were reared continuously via rice feeding (without exposure to any pesticides), which was replaced every 3–5 days under greenhouse conditions (27 °C ± 1 °C, 16:8 light/dark photoperiod, and 70%–80% relative humidity).

### 4.3. Preparation of Film-Forming Emulsion

The non-ionic surfactant Tween 20 was used as an emulsifier and solubilizer because of its excellent emulsifying performance to citral [47,48] and its ability to increase the solubility of GG [68]. Our preliminary experiments showed that the maximum safe amount of citral was 30% in mass ratios, and exceeding this amount could damage the rice seedlings, while the maximum solubility of GG was 40% in Tween 20-in-water emulsion (10% tween 20 + 50% water). Therefore, the citral and guar gum film *(GC film)* agent contained 40% citral and guar gum in mass ratios in the Tween 20-in-water emulsion, and the mass ratios of citral/GG were controlled at 3:1, 2:1, 1:1, 1:2, and 1:3 (*w*/*w*) to determine the best combination of citral and GG. The control *GG film* agent contains 10% tween 20, 20% GG, and 70% water in mass ratios. The required amount of tween 20 and water phases were mixed on an agitator for 30 min at room temperature; then, GG was slowly added to Tween 20-in-water emulsion, stirred for an hour, and ultrasonicated for another hour. Afterward, the required amount of citral was added to the emulsions and stirred for at least 1 h. Each film emulsion was freshly prepared immediately before use, for a total emulsion mass of 40 g.

### 4.4. Bioactivity Test of Film-Forming Emulsion

#### 4.4.1. Toxicity Bioassay

The toxicity of the film-forming emulsion to BPH was tested by the rice stem dipping method [8,69,70]. In brief, 200 mL of the citral (with 0.02% Tween 20), GG, and GC film emulsion agent liquid of each concentration was prepared. Rice seeds were germinated indoors. Afterward, fifteen 7 d-old rice seedlings were washed thoroughly, air-dried, and then grouped together and immersed into a concentration of film-forming emulsion for 30 s. The rice seedlings were air-dried at room temperature for at least 30 min, wrapped with water-impregnated cotton, and planted in 1000 mL plastic cups (20 cm diameter × 15 cm height) after they were air-dried at room temperature for at least 30 min. At last, fifteen third-instar nymphs were transferred onto a plastic cup that was bound with muslin cloth tied with a rubber band. Three replicates were obtained for each concentration, and one cup was treated as one replicate. Insects that were moribund or unable to move after a slight push with a fine brush were counted as dead. Statistical analysis of the toxicity data was performed using probit analysis to determine the 50% lethal concentration (LC_50_).

#### 4.4.2. Mortality Test

The mortality of BPH was studied using the spraying method reported by Yin et al. (2008) with slight modifications [71]. Four concentrations of each film-forming emulsion (100, 200, 400, and 800 dilution multiples) based on our previous experiments were prepared, and distilled water was set as the control. The rice seedlings were sprayed with 10 mL of liquid by a small manual sprayer with a cone nozzle (2 mm-diameter orifice). The treated seedlings were wrapped with water-impregnated cotton and planted in 1000 mL plastic cups. Then, fifteen third-instar nymphs and 1 d-old adults of BPH were released. All concentrations and controls were replicated three times. The number of BPH at different states were respectively counted and recorded after 60 h of growth in the conditions described above. Mortality was calculated using Abott’s formula.

### 4.5. Fourier-Transform Infrared (FTIR) Spectroscopy

The film-forming emulsion was frozen at a low temperature and dried in a desiccator for at least 24 h via a vacuum freeze dryer (LGJ-10E, Beijing, China). Samples were ground in an agate mortar containing KBr at room temperature prior to analysis. The FTIR spectra of the composite film were characterized on a Tensor 27 infrared microscope (Bruker, Ettlingen, Germany) over the wavenumber of 4000–400 cm^−1^ with a spectral resolution of 0.4 cm^−1^.

### 4.6. Preparation of Films

Alginate film-forming solutions were ultrasonically treated with a vacuum pump to remove air bubbles and avoid the presence of micro-holes in the film structure. Then, the films were formed by casting on glass plates with partitions of 30 × 40 cm. The films were dried at room temperature for 36–48 h and peeled off for further determination [72].

### 4.7. Microstructure of the Films

The figure of the films used digital still cameras (Canon and Nikon, Tokyo, Japan). The microstructure of the films was evaluated using a depth-of-field stereoscopic three-dimensional microscope, VHX-1000 (Keyence, Hong Kong). Scanning electron microscopy (SEM) (Hitach S3400N, Tokyo, Japan) of the films was performed under the following conditions: Voltage of 10 kV, magnification of 5000×, and at low vacuum of 60 Pa. Atomic force microscopy (AFM) was performed with a Dimension FastScanTM probe microscope (Bruker Biosciences Company, Ettlingen, Germany) and tested in knock mode with a silicon probe with a resonance frequency of 331.179 kHz, scanning range of 1 μm × 1 μm, and scanning frequency of 0.5 Hz.

### 4.8. Physicochemical Properties of the Films

Film thickness, taken as the average of five random locations, was obtained using a digital caliper micrometer (Thorlabs, NJ, USA) according to the method of Mei and Zhao (2003) [73]. Tensile properties and elongation at the break of the film were evaluated according to the method of Giteru et al. (2015) [74]. Tensile strength (TS, N/mm^2^) = F/A, where F (N) is the maximum force at break and A (mm^2^) is the initial cross-sectional area (thickness (mm) × width (mm)). Elongation at break (EAB, %) = (increase in length at breaking point (△l)/original length) × 100, where (△l) is the difference between the original distance among grips holding the specimen before and the distance after the break of the sample. The solubility of the films in water was determined by using a modification of the method of Kavoosi et al. (2013) [75]. Solubility (%) = (film initial dry weight - film final dry weight)/film initial dry weight × 100. Oxygen permeability was determined according to the method of Ayranci and Tunc (2003) [46]. Oxygen permeability (OP) = (m × d)/(A × t × △P), where m is the mass of oxygen (g) diffusing through the film; d is the thickness (m) of the film; A (m^2^) is the area through which permeation takes place; t (h) is the time interval during which permeation occurs; and △P (Pa) is the pressure difference of oxygen between the two sides of the film. The oxygen pressure on the nitrogen side of the film was assumed to be zero as all diffused oxygen was continuously carried away by nitrogen.

### 4.9. Statistical Analysis

All experiments were conducted three times with duplicate measurements of different samples. Statistical analyses were performed using student’s *t*-test and one-way ANOVA (*p* ≤ 0.05). Data were analyzed with Predictive Analytics Software (SPSS) for Windows version 18.0 (SPSS Inc., Chicago, IL, USA). The average nymph and adult mortality data were subjected to probit analysis to calculate the median lethal concentration (LC_50_), and other statistics at a 95% confidence limit and chi-square values were calculated to obtain the regression equation values.

## 5. Conclusion

A GC film that effectively repels BPH and is environmentally friendly was formulated in this work. The GC film with a citral/GG ratio of 1:1 (*w*/*w*) showed the best bioactivity to repel BPH, with the mortality rate ranging from 46.67% to 82.22% and from 37.78% to 71.11% in BPH adults and nymphs in the third instar, respectively, far below the LC_50_ of GG/citral individual. The mechanical properties of this film were not affected by citral addition, but the physical properties were affected. SEM and AFM results showed that GG had good compatibility with citral and was improved by the addition of essential oils. Our data indicate that the GG-based film incorporated with citral has good utilization potential in the new useful management of BPH due to its biodegradability and environmental friendliness. Further studies on improving the bioactivity of this film and clarifying its mechanism are still needed.

## Figures and Tables

**Figure 1 molecules-25-02044-f001:**
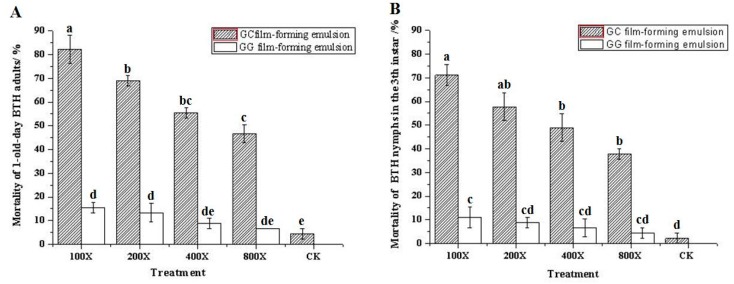
Mortality of (**A**) BPH adults and (**B**) BPH nymphs in the 3rd instar treated with different dilution multiples of film-forming emulsion. Value is mean ± SE (*n* = 3). The means followed by the same letter in the bar diagram are not significantly different according to ANOVA and Tukey’s multiple comparison tests (*p <* 0.05).

**Figure 2 molecules-25-02044-f002:**
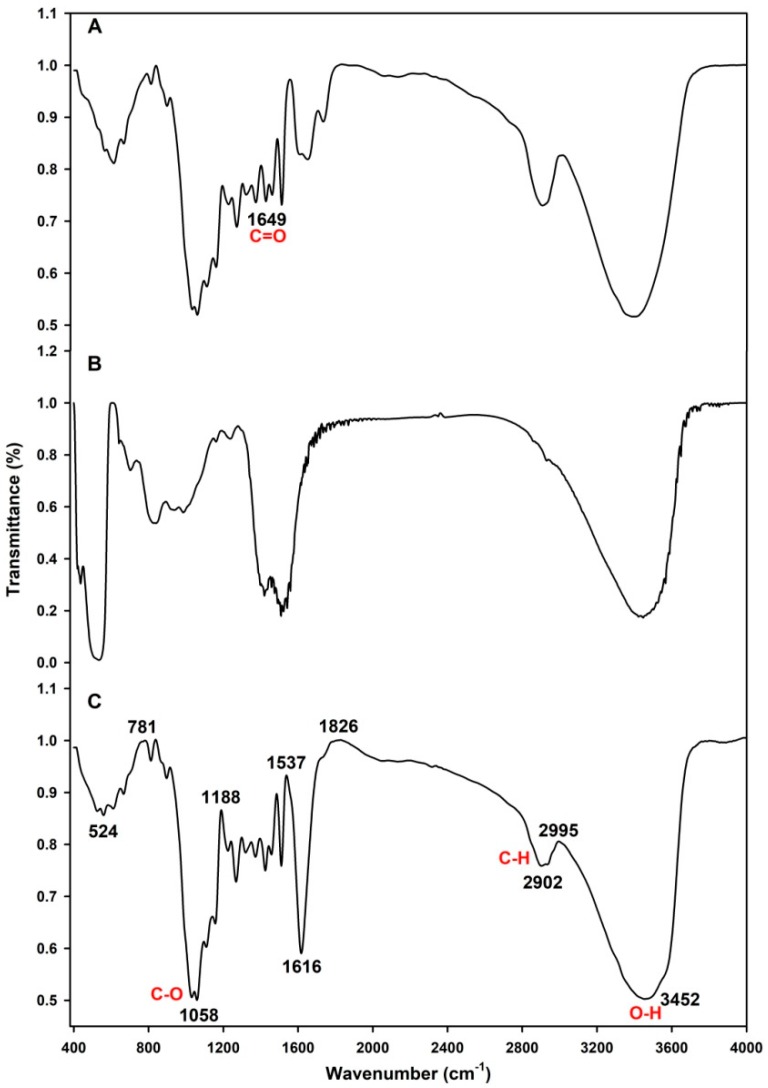
FTIR spectra of the (**A**) GC film-forming emulsion, (**B**) guar gum (GG) film-forming emulsion, and (**C**) pure guar gum**.**

**Figure 3 molecules-25-02044-f003:**
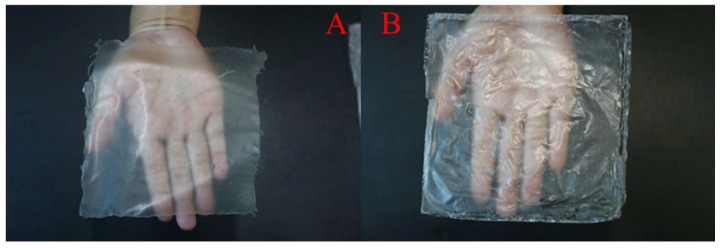
Visual observation of the different kinds of films. The figures were evaluated by a digital still camera (Canon and Nikon, Japan): (**A**) GC film, (**B**) GG film.

**Figure 4 molecules-25-02044-f004:**
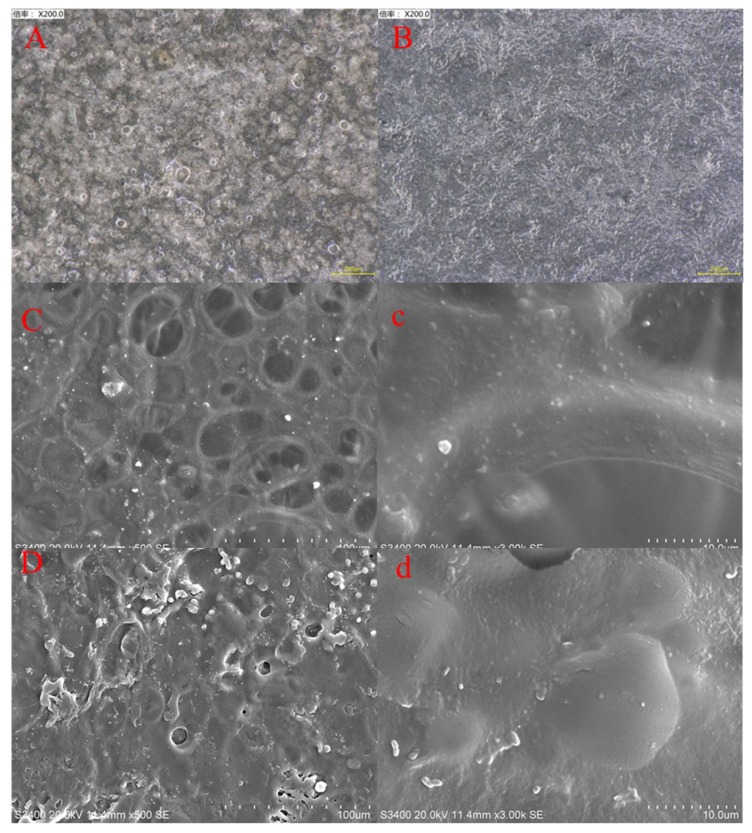
Microstructural features of the different types of films. Depth-of-field stereoscopic three-dimensional microscopy of (**A**) GC film and (**B**) GG film. SEM micrographs of films of (**C**,**c**) GC film and (**D**,**d**) GG film.

**Figure 5 molecules-25-02044-f005:**
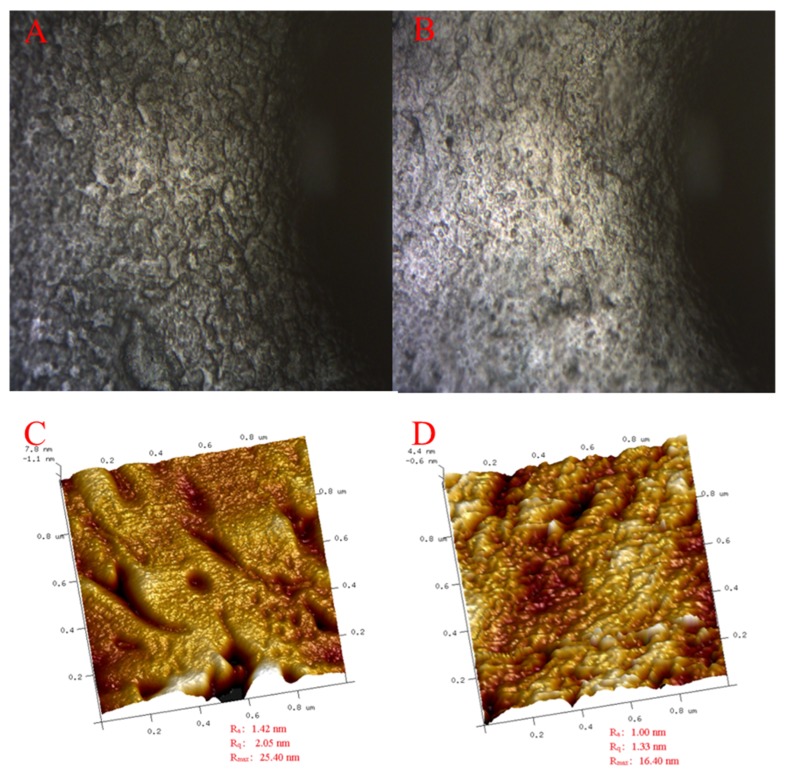
AFM images of the different types of films. The scanning scale is 1.0 × 1.0 μm. (**A**,**C**) GC film, H was planar and J was triaxial. (**B**,**D**) GG film, I was planar and K was triaxial.

**Table 1 molecules-25-02044-t001:** Median lethal concentration (LC_50_) of different ratios of citral and guar gum in GC film emulsions to 3rd-instar nymph of BPH.

Ratio of Citral and Guar Gum	Total Number of Nymphs Treated	Toxicity Regression Equations (Y)	Correlation Coefficient (r)	LC_50_ (95% Confidence Interval) (mg/mL)	Chi-Square Value
3:1	225	y = 4.262 + 0.981x	0.980	5.65 (2.58−12.39)	0.8397
2:1	225	y = 4.262 + 0.867x	0.992	7.11 (2.48−20.40)	0.2124
1:1	225	y = 4.475 + 0.828x	0.976	4.30 (1.64−11.31)	0.6948
1:2	225	y = 3.782 + 1.334x	0.998	8.19 (2.98−22.47)	0.0514
1:3	225	y = 4.014 + 0.830x	0.976	15.43 (3.12−76.21)	0.6044
citral	225	y = 3.769 + 0.748x	0.978	44.38 (2.06−954.28)	0.3503
GG	225	y = 3.483 + 0.705x	0.957	141.51 (3.72−5387.85)	0.6624

**Table 2 molecules-25-02044-t002:** The physicochemical properties of different types of films.

Film Type	Mechanical Properties			Physical Properties	
	Thickness (mm)	TS (N/mm^2^)	EAB (%)	SOW (%)	OP (×10^−9^cm^3^/m^2^·d·Pa)
GG film	39.10 ± 2.80 ^a^	41.89 ± 1.96 ^a^	1.82 ± 0.38 ^a^	30.30 ± 0.26 ^a^	8.26 ± 0.38 ^a^
GC film	41.70 ± 2.30 ^a^	38.30 ± 1.40 ^a^	2.03 ± 0.49 ^a^	15.00 ± 0.23 ^b^	2.37 ± 0.21 ^b^

Note: The values are displayed as mean ± SE (*n* = 5). Same superscript letters in the same column are not significantly different (*p* > 0.05). GG film: Guar gum film; GC film: Guar gum-based film incorporated with citral; TS: Tensile strength; EAB: Elongation at break; SOW: Solubility in water of film; OP: Oxygen permeability of film.

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
