# Peer review of "Physicochemical Properties and Bioactivity of a New Guar Gum-Based Film Incorporated with Citral to Brown Planthopper, Nilaparvata lugens (Stål) (Hemiptera: Delphacidae)"

_molecules, 2020, doi:10.3390/molecules25092044_

Round 1
Reviewer 1 Report
I have no objection to the publication.
The paper deals with GC film as a repellent agent against Nilaparta lugens.
The authors demonstrated the best citral and GG proportion showing the most
effective lethality. The authors observed GG-based film using SEM and AFM,
suggesting that the guar gum-based film is a good agent for the pest control.
Although I have no practical experience of FTIR and AF analyses,
both these studies seem to have been competently conducted.
And, the authors wrote and illustrated paper.
The study is a well written, and useful method.
Author Response
Thanks for your positive comments on our study and we will polish the manuscript further.
Reviewer 2 Report
The manuscript presents the result of a study investigating the effects of a guar-gum emulsion, eventually added with citral extracted from essential oil, against brown planthopper.
The manuscript is generally well written and organized, even if I found some minor spell check work that needs to be done.
The following concerns need to be clarified before considering this work for publication:
- The lethal / mortality effects recorded are based on bioassys in which rice seedlings were treated with GC or GG and exposed to insects. Then mortality of control is compared to mortality of different treated groups (different concentrations). However, these kinds of bioassays show an indirect effect on insect. For instance the gum/citral products may just have a repellent or antifeedant effect on insects that died by starvation; in such case it wuold not be possible to talk about direct toxicity. This is just an example to say that it would have been better to include in this study some test applying the insecticidal products directly on insects. This aspect is not secondary, because the purpose of this work appears to be the search for an alternative and environmentally friendly product, but the results are very preliminary and this study had to include more experiments to give some insight or explanation on the observed mortality. These aspects need to be discussed appropriately.
- I found difficult to compare results (especially in graphs) between GC and GG treatments, because of the use of these abbreviations “GC” and GG” that are very similar.
- This study includes two parts: bioassays on insects and physico-chemical analysis of the gum products. However it is not clear how these two parts link together for the observed effects on insects. This should be better discussed.
Author Response
Thank you for your comments concerning our manuscript entitled "Physicochemical properties and bioactivity of a new guar gum-based film incorporated with citral to brown planthopper, Nilaparvata lugens (Stål) (Hemiptera: Delphacidae)"( Manuscript ID: molecules-749890). Your comments are all valuable and very helpful for revising and improving our manuscript, as well as the important guiding significance to our researches. We have carefully studied comments and have made corrections which we hope meet with approval. The responds to your comments and the corrections in the revised manuscript are as flowing:
Comment 1: The manuscript is generally well written and organized, even if I found some minor spell check work that needs to be done.
Response: Thanks for your comments, I had revised some minor spell check work, the changes were highlighted in yellow.
Comment 2: The results are very preliminary and this study had to include more experiments to give some insight or explanation on the observed mortality. These aspects need to be discussed appropriately.
Response: Special thanks to you for your good comments! Rice stem dipping method is the most common method to test pesticide toxicity and As we know, citral has insecticidal activity, so we firstly designed our experiment based on this character. The data we obtained also supported this activity. Exactly as the reviewer said, the guar gum/citral products may just have a repellent or antifeedant effect on insects that died by starvation; in such case it wuold not be possible to talk about direct toxicity. We added a experiment to test the guar gum(GG)/citral individual of Median lethal concentration (LC50) to 3rd-instar nymph of Nilaparvata lugens. The data were added on Table 1 . These data show an direct toxicity of our emulsion on Nilaparvata lugens, for the LC50 of guar gum(GG)/citral individual were extremely High than the emulsion.
Besides, combined with the Figure 1.Mortality of (A) BTH adults and (B) BTH nymphs in the 3th instar treated with different dilution multiples of film-forming emulsion. The GG film-forming emulsion could had a certain effect on mortality of BTH only in high concentration(100X-200X) and these effect were very lowed compared with GC film-forming emulsion in the same GG concentration. So the observed mortality most dued to insecticidal activity of film-forming emulsion, and a few dued to have a repellent or antifeedant effect.
Correspondingly, the explanation on the observed mortality were shown in 2. Results(See lines 111-114,125-127) and discussed in 3.Discussion(See lines198-199,215), highlighted in yellow.
Comment 3: I found difficult to compare results (especially in graphs) between GC and GG treatments, because of the use of these abbreviations “GC” and GG” that are very similar.
Response: The abbreviations “GG” was the guar gum, it is a common abbreviation and appeared in many articles. The abbreviations “GC” was the guar gum+citral, we used this abbreviation to clearly indicated the component of emulsion. To avoid misunderstanding, we also clearly revised the meaning of these abbreviations “GC” and GG” in manuscript, highlighted in yellow(Line 69 and line 107).
Comment 4: This study includes two parts: bioassays on insects and physico-chemical analysis of the gum products. However it is not clear how these two parts link together for the observed effects on insects. This should be better discussed.
Response: Thanks very much for the your good advice, we futher dicussed this part in the 3.Discussion(Line 237-239,276-280).
Reviewer 3 Report
The manuscript fits within the scope of the journal and the authors have knowledge and expertise in this field of study. However, the idea of this work is not particularly original. Moreover, several improvements are still needed in order to achieve a suitable manuscript:
- I have observed that the authors are writing the manuscript using very long sentences. It would be better if they shorten these sentences to be clearer and simpler for readers. Moreover, the English grammar should be improved.
- A concise and factual abstract is required. The abstract should state briefly the purpose of the research, the principal results and major message.
- It might be useful to deepen some cases of Integrated Pest Management (IPM) practices, such as those promoted by the European Union (see for example “Studies in Natural Products Chemistry 2014, 43, 437-482”, “Phil Trans R Soc: Biol Sci. 2011, 366, 1987-1998”, etc.), and the potential of this new guar gum-based film within the IPM practices.
- In the current work, variability represents a concern for this reviewer. The authors should explain in more detail how this concern was controlled/studied in sample preparation steps.
- In general, it is possible to describe in more detail the results with detailed mathematical refinement.
- It should also be provided a quantitative and believable estimation of the necessities that could be covered by the proposed approach.
- References need to be updated. All references before 2010 should be removed unless they are essential for understanding the manuscript.
Taking into account the comments to the authors, I consider that major revisions are still needed before the manuscript could be accepted in this journal. However, this reviewer encourages the authors to improve the manuscript and address the issues commented above, because the work is of interest. After an appropriate revision, I think that the manuscript could be of interest for Molecules.
Author Response
Thank you for your comments concerning our manuscript entitled "Physicochemical properties and bioactivity of a new guar gum-based film incorporated with citral to brown planthopper, Nilaparvata lugens (Stål) (Hemiptera: Delphacidae)"( Manuscript ID: molecules-749890). Your comments are all valuable and very helpful for revising and improving our manuscript, as well as the important guiding significance to our researches. We have carefully studied comments and have made corrections which we hope meet with approval. The responds to your comments and the corrections in the revised manuscript are as flowing:
Comment 1: I have observed that the authors are writing the manuscript using very long sentences. It would be better if they shorten these sentences to be clearer and simpler for readers. Moreover, the English grammar should be improved.
Response: Thanks for your comments, we agree with this suggestion and had revised the English grammar and long sentences according to your comments in the attached file, the changes were highlighted in yellow.
Comment 2: A concise and factual abstract is required. The abstract should state briefly the purpose of the research, the principal results and major message.
Response: Thanks for your comment, we had revised abstract on both language and expression according to your comments in the attached file, and involved a professor who is good at English writing for language corrections. The changes were highlighted in yellow.
Comment 3: It might be useful to deepen some cases of Integrated Pest Management (IPM) practices, such as those promoted by the European Union (see for example “Studies in Natural Products Chemistry 2014, 43, 437-482”, “Phil Trans R Soc: Biol Sci. 2011, 366, 1987-1998”, etc.), and the potential of this new guar gum-based film within the IPM practices.
Response: Special thanks to you for your good comments! we added the discussion of the potential of this new guar gum-based film within the IPM practices in the 3.Discussion(Line 274 to line 280) and added three references:
Regulation (EC) No. 1107/2009, Off. J. Eur. Union L. 309, 2009, 1-50.
Chandler, D.; Bailey, A.S.; Tatchell, G.M.; Davidson, G.; Greaves, J.; Grant, W.P. The development, regulation and use of biopesticides for integrated pest management. Philosophical Transactions of the Royal Society B: Biological Sciences, 2011, 366(1573), 1987-1998. DOI:10.1098/rstb.2010.0390.
Villaverde, J.J.; Sevilla-Morán, B.; Sandín-España, P.; López-Goti, C.; Alonso-Prados, J.L. Challenges of biopesticides under the European regulation (EC) No. 1107/2009. Studies in Natural Products Chemistry, 2014, 43,437-482. DOI:10.1016/b978-0-444-63430-6.00015-1.
Comment 4: In the current work, variability represents a concern for this reviewer. The authors should explain in more detail how this concern was controlled/studied in sample preparation steps.
Response: Thanks for your comment, we described in detail in sample preparation steps in 4.3. Preparation of film-forming emulsion (See lines 306-310).
Comment 5: In general, it is possible to describe in more detail the results with detailed mathematical refinement.
Response: Thanks for your comment, I had rewrited the 2.0 Results (See lines 111-114,125-127,150-152,159-160,162-164) you mentioned and highlighted in yellow.
Comment 6: It should also be provided a quantitative and believable estimation of the necessities that could be covered by the proposed approach.
Response: Thanks for pointing out this issue, we indeed should have provided a quantitative and believable estimation of the necessities. We had rewrited you mentioned and highlighted in yellow(See lines 306-310,314-315).
Comment 7: References need to be updated. All references before 2010 should be removed unless they are essential for understanding the manuscript.
Response: Accepted. Some references as follows were removed for consensus, some references before 2010 were not removed for understanding or method.
[20]Barth, H.G.; Smith, D.A. High-performance size-exclusion chromatography of guar gum. J Chromatogr, 1981, 206, 410-415. DOI:10.1016/S0021-9673(00)82558-3.
[21]Vijayendran, B.R.; Bone, T . Absolute molecular weight and molecular weight distribution of guar by size exclusion chromatography and low-angle laser light scattering. Carbohydr Polym.1984, 4, 299-311. DOI:10.1016/0144-8617(84)90005-5.
[44]Abbott, W.S. A methodology of computing the effectiveness of an insecticide. J. Econ. Entomol, 1925, 18, 265-267. DOI:10.1093/jee/18.2.265a.
[50]Finney, D.J. Probit analysis (3rd ed.). Cambridge: Cambridge University press, 1971.
Round 2
Reviewer 2 Report
The authors have improved the manuscript according to reviewer's comments.
The manuscript still needs some spell-check work before publication.
Reviewer 3 Report
In my judgment, after review carefully the manuscript, the changes are adequate and the authors have improved it properly. The confusing issues are also clearer. Currently, the article includes all the necessary information for a proper understanding of the work and results are of interest.